# Efficacy of Novel Combinations of Antibiotics against Multidrug-Resistant—New Delhi Metallo-Beta-Lactamase-Producing Strains of *Enterobacterales*

**DOI:** 10.3390/antibiotics12071134

**Published:** 2023-06-30

**Authors:** Shamsi Khalid, Antonella Migliaccio, Raffaele Zarrilli, Asad U. Khan

**Affiliations:** 1Medical Microbiology Laboratory, Interdisciplinary Biotechnology Unit, Aligarh Muslim University, Aligarh 202001, India; shamsi123khalid@gmail.com; 2Department of Public Health, University of Naples Federico II, 80131 Naples, Italy; antonella.migliaccio10@gmail.com

**Keywords:** antimicrobial resistance, antibiotics synergy, carbapenemase, NDM-beta-lactamase

## Abstract

The emergence of multidrug-resistance (MDR)—New Delhi metallo-beta-lactamase (NDM)-producing microorganisms—has become a serious concern for treating such infections. Therefore, we investigated the effective antimicrobial combinations against multidrug-resistant New Delhi metallo-beta-lactamase-producing strains of *Enterobacterales*. The tests were carried out using the 2D(two-dimensional) checkerboard method. Of 7 antimicrobials, i.e., doripenem (DRP), streptomycin (STR), cefoxitin (FOX), imipenem (IPM), cefotaxime (CTX), meropenem (MER), and gentamicin (GEN), 19 different combinations were used, and out of them, three combinations showed synergistic effects against 31 highly drug-resistant strains carrying *bla_NDM_* and other associated resistance markers. Changes in the minimum inhibitory concentration (MIC) values were interpreted using the test fractional inhibitory concentration index (FIC Index). The FIC Index values of these combinations were found in the range of 0.1562 to 0.5, which shows synergy, whereas no synergism was observed in the remaining antimicrobial combinations. We conclude that these antibiotic combinations can be analyzed in in vivo and pharmacological studies to establish an effective therapeutic approach.

## 1. Introduction

The emergence of multidrug-resistant (MDR) bacterial infections is one of the major worldwide problems having difficulty being treated. Metallo-β-lactamases (MBLs) and extended β- lactamases (ESBLs) are the major causes of resistance in bacteria against available antibiotics [1]. Antibiotic efficacy against many Gram-negative pathogens is increasingly compromised by the spread and emergence of MDR strains that produce β-lactamases. These pathogens can confer resistance to one or more carbapenems, cephalosporins, monobactam penicillin, or drugs that are routinely used in clinical practice [2]. Carbapenems are considered the last resort of antibiotics against infections caused by MDR gram-negative microorganisms that show stability and high resistance values through bacterial outer membranes [1]. The common mechanisms for the members of the Enterobacteriaceae family include the production of β- lactamases, efflux pumps, and modification of penicillin-binding proteins (PBPs). In some bacterial species, the combination of these pathways can result in significant resistance to carbapenems [3]. In contrast to monobactam, New Delhi Metallo- β-lactamases (NDM) are a form of MBL that can hydrolyze most β-lactams, including carbapenems. The primary antimicrobial medicines of choice for treating severe infections caused by Gram-negative bacteria are carbapenems. Clinically used β-lactamase inhibitors, such as avibactam, clavulanate, sulbactam, and tazobactam, cannot stop the hydrolysis of β-lactams by NDM enzymes. In 2008, a Swedish patient who had been hospitalized in New Delhi, India, was infected with a strain of *Klebsiella pneumoniae* that included NDM-1 for the first time [4]. Since then, NDM-1 has been detected in numerous *Enterobacteriaceae*, *Acinetobacter*, and *Pseudomonas* species. These infections are the major challenge for clinicians in treating critically ill patients. The risk factors for NDM-1 strains include the lack of effective antibiotics, the failure to recognize highly prevalent asymptomatic carriers, the absence of a routine phenotypic test for the detection of Metallo-beta lactamases, and the presence of MBL on plasmids with the potential to rearrange and spread through horizontal gene transfer [3]. Therefore, to overcome the resistance problem, several studies for the treatment of MDR bacterial infections can be observed with the combination of ≥2 antimicrobial agents [5,6]. In the current scenario of the emergence of antibiotic resistance, where all classes of antibiotics fail to treat infections, a combination of these available antibiotics may be one of the most intelligent ideas to subside infections. Antimicrobial combination therapies can increase their efficacy over monotherapy and decrease MDR-based infections in clinical settings. Hence, we proposed to screen a large number of combinations of these available antibiotics against a set of MDR strains of clinical origin that have already been characterized in our previous studies.

## 2. Results and Discussion

### 2.1. Minimum Inhibitory Concentration

The MICs of cefoxitin, doripenem, imipenem, and streptomycin against MDR strains carrying blaNDM and other associated resistance markers on a plasmid, were reported in the range between 4096 μg/mL and 128 μg/mL (Table 1). All these MDR strains have already been characterized for the presence of resistance markers in our laboratory, and these isolates were collected from the neonatal intensive care unit (NICU) of an Indian hospital [7,8,9,10,11,12,13].

### 2.2. Synergistic Effect of Antibiotic Combinations

MDR clinical strains harboring *blaNDM* and other associated resistance markers like *blaOXA-1*, *blaCTX-M*, *blaAmpC*, *blaCMY-1*, and *blaSHV* showed resistance toward doripenem (MIC ranges 256 μg/mL to 1024 μg/mL), imipenem (MIC ranges 128 μg/mL to 2048 μg/mL), cefoxitin (MIC ranges 256 μg/mL to 4096 μg/mL) and streptomycin (MIC ranges 512 μg/mL to 4096 μg/mL) which were previously well characterized in our laboratory (Table 1). To check the effect of these antibiotics in combination, a total of 19 combinations of different classes of antibiotics were tested by a 2D checkerboard microdilution assay. Of them, only three combinations were found to be most effective against the set of clinical strains (Table 1). These combinations belong to the classes of carbapenems, cephamycin, and aminoglycosides (doripenem with cefoxitin, doripenem with streptomycin, and imipenem with cefoxitin), showing synergy. It was observed that the MICs of doripenem decreased from 1024 μg/mL to 64 μg/mL, cefoxitin 4096 μg/mL to 64 μg/mL, and streptomycin 4096 μg/mL to 32 μg/mL (Table 1) in combination, and their FICI values were in the range of 0.156 to 0.5 (Table 1) for all highly resistant strains tested, which were found in synergistic range. Previously, a synergistic effect of doripenem in combination with cefoxitin and tetracycline in inhibiting *blaNDM-1*-producing bacterial strains was reported by our research group [14]. The synergistic interaction of antimicrobials allows the use of lower doses. Another combination of imipenem and cefoxitin was showing a synergistic effect, with FICI values in the range of 0.187 to 0.5 (Table 1). It has been previously reported that synergy is observed when the FICI value is ≤0.5 [14]. In this study, three combinations showed synergy. The FICI values of combinations were found to be 0.5 to 0.1562 (Table 1). Although several mechanism-based studies were performed earlier on the interaction of antibiotics with specific markers using biophysical and biochemical approaches [14,15,16]. Antibiotic combination therapy with ceftazidime/avibactam (CAZ/AVI) and aztreonam (ATM) was previously investigated for the treatment of infection with NDM producer Enterobacterales. The majority of Enterobacterales that are ATM resistant and NDM positive shows significant efficacies of the CAZ/AVI+ATM combination against them [17]. Another study demonstrated the effectiveness of the double carbapenem combination against Gram-negative bacteria through the in vitro synergistic activity of ertapenem and meropenem [18].

Although β-lactamase inhibitors have been crucial to fighting against β-lactam resistance in Gram-negative bacteria, their potency has been dwindling as a result of the development of numerous severe β-lactamases. Though it has a unique synergistic mechanism of action, a triple combination of β- lactam antibiotics meropenem, piperacillin, and tazobactam has been demonstrated to be an effective method for killing Methicillin-resistant *Staphylococcus aureus* (MRSA) in vitro and in a mouse model [19]. 

The susceptibility pattern of *bla_NDM-_*producing bacteria may vary geographically depending on specific strains over time. In this study, all microorganisms present were NDM producers and showed high resistance values due to which synergy of antibiotics was not able to restore susceptibilities, but the combination values were in synergistic range, showing this combination can be used for the therapeutic approach.

## 3. Materials and Methods

### 3.1. Strain, Antibiotics and Chemicals

This study included 31 MDR clinical strains carrying blaNDM and other associated resistance markers to determine the minimum inhibitory concentration (MIC) and fractional inhibitory concentration index (FICI). These strains were obtained from NICU of a North Indian Hospital. These are ESBL- and MBL-producing strains with different resistant markers, as reported previously by our group [7,8,9,10,11,12,13]. The antibiotic resistance markers and MIC of these strains are presented in Table 1. Doripenem, cefoxitin, and imipenem were purchased from Sigma-Aldrich (Sigma, Milan, Italy). Streptomycin was purchased from Himedia (Mumbai, India). Mueller–Hinton broth was purchased from Himedia (Mumbai, India).

### 3.2. Combination of Antibiotics and MIC

The antibiotics used in this study were doripenem (DRP), streptomycin (STR), cefoxitin (FOX), imipenem (IPM), cefotaxime (CTX), meropenem (MER), and gentamicin (GEN). These antibiotics were used to prepare all possible combinations against highly resistant clinical strains. To examine the MIC of antibiotics for a set of clinical strains, the overnight-grown colonies were collected using a sterile loop and transferred into a tube containing 5 mL of Mueller–Hinton broth. This broth was incubated at 37 °C to obtain a final turbidity equivalent to that of 0.5 McFarland standards (10^8^ CFU/mL) and diluted to 1:100 for the broth microdilution procedure. The strains were treated with decreasing drug concentrations from 4096 μg/mL to 0.5 μg/mL according to Clinical Laboratory Standards Institute (CLSI) guidelines [20].

### 3.3. D-Checkerboard Microdilution Assay to Determine FICI

A total of 19 different combinations of antibiotics were taken against 31 MDR clinical strains. A 2D checkerboard microdilution assay was performed using a 96-well microtiter plate. Serially diluted antibiotics were taken in concentrations less than, equal to, or greater than their MICs. To check their effect against MDR clinical stains, fractional inhibitory concentration indexes (FICI) were calculated. The FICI value ≤0.5 was defined as synergy, <4, indifference, and >4, antagonism [21].

## 4. Conclusions

The study revealed three combinations: doripenem with cefoxitin, doripenem with streptomycin, and imipenem with cefoxitin, which are effective against highly drug-resistant clinical strains carrying blaNDM and other associated resistance markers. Hence, we propose these novel combinations against highly drug resistant clinical strains of bacteria for further in vivo and pharmacological studies in order to establish effective infection control therapy.

## Figures and Tables

**Table 1 antibiotics-12-01134-t001:** Minimum Inhibitory concentrations and synergistic effects of doripenem, cefoxitin, imipenem, and streptomycin antibiotics against 31 MDR-NDM-producing clinical strains.

Strains	Resistance Markers	MIC µg/mL				MIC µg/mL FIC Index ^5^			Ref.
		DRP ^1^	FOX ^2^	IMP ^3^	STP ^4^	DRP + FOX	DRP + STP	IMP + FOX	
AK-33 *Escherichia coli*	NDM-4, OXA-1, CTX-M, Amp C	512	2048	1024	2048	128 + 512 0.5	128 + 256 0.375	128 + 256 0.25	[13]
AK-35 *Escherichia coli*	NDM-7, OXA-1	1024	2048	1024	2048	128 + 256 0.25	128 + 128 0.1875	256 + 512 0.5	[13]
AK-37 *Escherichia coli*	NDM-1, CMY-139, OXA-1, CTX-M	1024	1024	512	512	128 + 128 0.375	256 + 128 0.5	256 + 128 0.5	[13]
AK-83 *Escherichia coli*	NDM-7, OXA-1, SHV-1	512	4096	1024	2048	64 + 512 0.25	128 + 512 0.5	128 + 512 0.25	[8]
AK-66 *Klebsiella pneumoniae*	NDM-1, OXA-1, OXA-9, CMY-1	256	1024	1024	2048	128 + 256 0.375	256 + 256 0.25	128 + 512 0.1875	[8]
AK-102 *Klebsiella pneumoniae*	NDM-5, OXA-1, OXA-9, CMY-4	1024	2048	1024	2048	128 + 512 0.375	64 + 512 0.3125	64 + 256 0.1875	[8]
AK-121 *Klebsiella pneumoniae*	NDM-1	512	1024	512	1024	128 + 256 0.5	64 + 256 0.375	128 + 256 0.5	[10]
AK-125 *Klebsiella pneumoniae*	NDM-1	512	1024	1024	2048	128 + 256 0.5	64 + 256 0.25	128 + 256 0.375	[10]
AK-130 *Klebsiella pneumoniae*	NDM-1	512	256	512	1024	64 + 64 0.375	128 + 128 0.375	128 + 64 0.5	[10]
AK-140 *Klebsiella pneumoniae*	NDM-1, OXA-48	1024	1024	2048	2048	128 + 256 0.375	256 + 512 0.5	512 + 128 0.375	[10]
AK-142 *Klebsiella pneumoniae*	NDM-1	512	1024	512	2048	128 + 526 0.5	64 + 512 0.375	128 + 128 0.25	[10]
AK-144 *Klebsiella pneumoniae*	NDM-1	512	2048	1024	1024	128 + 512 0.25	64 + 128 0.25	512 + 512 0.25	[10]
AK-147 *Klebsiella pneumoniae*	NDM-1, OXA-48	1024	2048	2048	1024	128 + 512 0.375	64 + 128 0.1875	512 + 512 0.5	[10]
AK-149 *Klebsiella pneumoniae*	NDM-1, OXA-48	1024	2048	2048	2048	128 + 512 0.375	512 + 128 0.375	256 + 512 0.375	[10]
AK-158 *Klebsiella pneumoniae*	NDM-5	1024	2048	2048	2048	128 + 256 0.25	64 + 512 0.3125	512 + 512 0.5	[10]
AK-100 *Klebsiella oxytoca*	NDM-4, OXA-1, OXA-9	1024	4096	1024	2048	256 + 256 0.3125	256 + 512 0.5	128 + 512 0.25	[8]
AK-67 *Enterobacter aerogenes*	NDM-1, OXA-1, SHV-2	1024	2048	1024	2048	64 + 128 0.1875	128 + 64 0.1562	32 + 64 0.3125	[8]
AK-93 *Enterobacter aerogenes*	NDM-4, OXA-1, OXA-9, SHV-1	512	1024	256	1024	64 + 64 0.185	128 + 128 0.375	32 + 64 0.25	[11]
AK-95 *Enterobacter aerogenes*	NDM-5, OXA-1, OXA-9, CMY-149	256	1024	256	1024	64 + 128 0.375	64 + 256 0.5	32 + 256 0.375	[11]
AK-96 *Enterobacter aerogenes*	NDM-7, OXA-1, OXA-9, CMY-145	256	1024	256	2048	64 + 128 0.375	32 + 256 0.375	64 + 256 0.5	[11]
AK-108 *Enterobacter cloacae*	NDM-4, OXA-1, OXA-9, CMY-149	512	2048	512	1024	128 + 256 0.375	128 + 256 0.5	64 + 256 0.25	[8]
AK-154 *Acinetobacter baumannii*	NDM-5	1024	2048	1024	2048	128 + 256 0.25	128 + 256 0.25	64 + 256 0.1875	[9]
AK-42 *Citrobacter freundii*	NDM-1, CMY-42, OXA-1, CTX-M, AmpC	1024	4096	1024	4096	256 + 1024 0.5	256 + 512 0.375	128 + 1024 0.375	[13]
AK-58 *Citrobacter freudii*	NDM-7, CMY-2, OXA-1, CTX-M	256	1024	128	2048	32 + 256 0.375	64 + 512 0.5	32 + 128 0.375	[13]
AK-82 *Citrobacter freundii*	NDM-4, OXA-9, SHV-1, CMY-149	512	4096	2048	2048	128 + 1024 0.5	64 + 256 0.25	128 + 512 0.1875	[8]
AK-48 *Citrobacter braakii*	NDM-4, CMY-4, OXA-48	512	1024	1024	4096	128 + 256 0.5	128 + 512 0.375	128 + 128 0.25	[13]
AK-49 *Citrobacter farmer*	NDM-4, CMY-4, OXA-48	256	1024	1024	2048	64 + 256 0.5	128 + 1024 0.5	128 + 128 0.25	[13]
AK-68 *Cedecea lapagei*	NDM-1, CTX-M, SHV, TEM	512	1024	512	1024	128 + 256 0.5	128 + 256 0.5	128 + 128 0.375	[12]
AK-152 *Cedecea davisae*	NDM-1	1024	2048	1024	1024	128 + 256 0.25	128 + 256 0.375	64 + 256 0.1875	[9]
AK-65 *Shigella boydii*	NDM-5, CMY-42, OXA-1, CTX-M	1024	2048	128	2048	64 + 64 0.125	128 + 128 0.1875	32 + 128 0.3125	[13]
AK-92 *Moellerella wisconsensis*	NDM-1	512	1024	256	2048	128 + 256 0.5	64 + 256 0.25	64 + 128 0.375	[7]

^1^ DRP—doripenem; ^2^ FOX—cefoxitin; ^3^ STP—streptomycin; ^4^ I MP—imipenem; ^5^ FIC Index—Fractional Inhibitory Concentration Index.

## Data Availability

All data are provided in the main text of the manuscript.

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
