# Peer review of "Efficacy of Novel Combinations of Antibiotics against Multidrug-Resistant—New Delhi Metallo-Beta-Lactamase-Producing Strains of Enterobacterales"

_antibiotics, 2023, doi:10.3390/antibiotics12071134_

Round 1

Reviewer 1 Report

The authors present a study looking at the combinations of antibiotics on MDR organisms. 

I would like the authors to add a small section to the paragraph where there have been clinical studies in vivo of using combination antibiotics or how these combinations can be included in treatment guidelines. I'm not aware how EUCAST or CLSI or other antibiotic guidelines have used combinations. 

Otherwise, the paper is clear, the conclusions are supported by the data and all information needed in the methods is included. 

Author Response

Reviewer 1:

I would like the authors to add a small section to the paragraph where there have been clinical studies in vivo of using combination antibiotics or how these combinations can be included in treatment guidelines. I'm not aware how EUCAST or CLSI or other antibiotic guidelines have used combinations. 

Our Response:

I appreciate the honourable reviewer’s comment to add small section on in vivo test of such combinations. Yes, we have added a little paragraph in the results and discussion section and highlighted. Not much studies have been reported and it is beyond EUCAST/CLSI guideline since varying combination have been reported earlier from other labs and from my own lab which need to be implemented in vivo model for translating to clinical trials.

We are keen to proceed further standardisation of these combination to treat animal models first and translate toward human trails in next studies. Therefore, this study is very significant to move further.

Reviewer 2 Report

This paper conducted the effective antimicrobial combinations against multidrug-resistant- New Delhi metallo-beta-lactamase-producing strains of Enterobacterales.

This paper used few bacterias and did just combinations effect.

So, they should conduct more experiment.

Good english

Author Response

As per Editor's suggestion, we addressed queries of reviewers 1 and 3.

Reviewer 1:

I would like the authors to add a small section to the paragraph where there have been clinical studies in vivo of using combination antibiotics or how these combinations can be included in treatment guidelines. I'm not aware how EUCAST or CLSI or other antibiotic guidelines have used combinations. 

Our Response:

I appreciate the honourable reviewer’s comment to add small section on in vivo test of such combinations. Yes, we have added a little paragraph in the results and discussion section and highlighted. Not much studies have been reported and it is beyond EUCAST/CLSI guideline since varying combination have been reported earlier from other labs and from my own lab which need to be implemented in vivo model for translating to clinical trials.

We are keen to proceed further standardisation of these combination to treat animal models first and translate toward human trails in next study. Therefore, this study is very significant to move further.

Reviewer 3:                                                       

  1. The big limitation of the study is no information about susceptibility to antibiotics of 31 of analysed strains. Please add this to the study e.g. as supplementary material.

Response: Isolates included in the study, have been characterized already in previous study which are all listed in the table and mentioned in Results and discussion section with their references with highlighted.

  1. Line 16 The Authors write “highly drug resistant”. What does it mean – multidrug-resistant, extensively drug-resistant or pandrug-resistant? We have world definitions for MDR, XDR or PDR phenotypes. Please explain.

Response: We mentioned “highly drug resistant” because the MICs of these antibiotics used against the resistance strains were very high. Though these are MDR strains.

We want to thank the reviewers once more for taking the time to read and comment on our article Efficacy of Novel Combinations of Antibiotics against Multidrug-Resistant- New Delhi Metallo-Beta-lactamase-Producing Strains of Enterobacterales. 

Reviewer 3 Report

The manuscript of Khalid et al. “Efficiacy of novel ...” is interesting, but needs some corrections.

1.       The big limitation of the study is no information about susceptibility to antibiotics of 31 of analysed strains. Please add this to the study e.g. as supplementary material.

2.       Line 16 The Authors write “highly drug resistant”. What does it mean – multidrug-resistant, extensivelydrug-resistant or pandrug-resistant? We have world definitions for MDR, XDR or PDR phenotypes. Please explain.

Author Response

Reviewer 3:                                                       

  1. The big limitation of the study is no information about susceptibility to antibiotics of 31 of analysed strains. Please add this to the study e.g. as supplementary material.

Response: Isolates included in the study, have been characterized already in previous study which are all listed in the table and mentioned in Results and discussion section with their references with highlighted.

  1. Line 16 The Authors write “highly drug resistant”. What does it mean – multidrug-resistant, extensively drug-resistant or pandrug-resistant? We have world definitions for MDR, XDR or PDR phenotypes. Please explain.

Response: We mentioned “highly drug resistant” because the MICs of these antibiotics used against the resistance strains were very high. Though these are MDR strains.

Round 2

Reviewer 2 Report

이 논문은 여전히 ​​많은 문제가 있습니다.  따라서 이 논문은 불합격 처리되어야 합니다.

.